# Extraction and Purification of Flavonoids from *Buddleja officinalis* Maxim and Their Attenuation of H_2_O_2_-Induced Cell Injury by Modulating Oxidative Stress and Autophagy

**DOI:** 10.3390/molecules27248985

**Published:** 2022-12-16

**Authors:** Shaofeng Wei, Xiaoyi Liu, K. M. Faridul Hasan, Yang Peng, Jiao Xie, Shuai Chen, Qibing Zeng, Peng Luo

**Affiliations:** 1School of Public Health, The Key Laboratory of Environmental Pollution Monitoring and Disease Control, Ministry of Education, Guizhou Medical University, Guiyang 550025, China; 2Simonyi Károly Faculty of Engineering, University of Sopron, 9400 Sopron, Hungary; 3Department of Biochemistry and Molecular Biology, College of Basic Medical Sciences, Guizhou Medical University, Guiyang 550025, China

**Keywords:** *Buddleja officinalis* Maxim, flavonoids, extraction, oxidative stress, autophagy

## Abstract

Cataracts are an ailment representing the leading cause of blindness in the world. The pathogenesis of cataracts is not clear, and there is no effective treatment. An increasing amount of evidence shows that oxidative stress and autophagy in lens epithelial cells play a key role in the occurrence and development of cataracts. *Buddleja officinalis* Maxim flavonoids (BMF) are natural antioxidants and regulators that present anti-inflammatory and anti-tumor effects, among others. In this study, we optimized the extraction method of BMFs and detected three of their main active monomers (luteolin, apigenin, and acacetin). In addition, a model of oxidative damage model using rabbit lens epithelial cells induced by hydrogen peroxide (H_2_O_2_). By detecting the levels of superoxide dismutase (SOD), glutathione peroxidase (GSH-Px), malondialdehyde (MDA), and OH (OH), the expression of autophagosomes and autolysosomes were observed after MRFP-GFP-LC3 adenovirus was introduced into the cells. Western blotting was used to detect the expression of Beclin-1 and P62. Our research results showed that the optimal extraction parameters to obtain the highest yield of total flavonoids were a liquid–solid ratio of 1:31 g/mL, an ethanol volume fraction of 67%, an extraction time of 2.6 h, and an extraction temperature of 58 °C. Moreover, the content of luteolin was 690.85 ppb, that of apigenin was 114.91 ppb, and the content of acacetin was 5.617 ppb. After oxidative damage was induced by H_2_O_2_, the cell survival rate decreased significantly. BMFs could increase the levels of superoxide dismutase (SOD) and glutathione peroxidase (GSH-Px) and decrease the levels of malondialdehyde (MDA) and OH (OH). After the MRFP-GFP-LC3 virus was introduced into rabbit lens epithelial cells and detecting the expression of P62 and Beclin-1, we found that the intervention of BMF could promote the binding of autophagosomes to lysosomes. Compared with the model group, the level of P62 in the low-, middle-, and high-dose groups of BMF was significantly down-regulated, the level of Beclin-1 was significantly increased, and the difference was statistically significant (*p* < 0.05). In other words, the optimized extraction method was better than others, and the purified BMF contained three main active monomers (luteolin, apigenin, and acacetin). In addition, BMFs could ameliorate the H_2_O_2_-induced oxidative damage to rabbit lens cells by promoting autophagy and regulating the level of antioxidation.

## 1. Introduction

Cataracts are the leading cause of blindness globally; it was estimated that more than 35 million people worldwide suffer from moderate or greater visual impairment, of which more than 10 million are caused by cataracts [1]. According to the World Health Organization, more than 40% of blindness cases are caused by cataracts [2], and nearly 90% of blindness is found in developed countries [3]. According to the literature, as early as 2008, the direct medical cost of cataracts in the United States was about USD 6.8 billion per year [3]. The high prevalence of cataracts has also occurred in other countries as well, resulting in a huge global economic burden.

The occurrence and development of cataracts is complex. Oxidative stress-induced autophagy in lens epithelial cells is considered to be one of the main mechanisms of cataract development [4]. Studies have shown that oxidative stress can cause lipid peroxidation in cells, causing the content of ATP (adenosine triphosphate) and Ca^2+^ in mitochondria to decrease. This, in turn, leads to mitochondrial swelling, destroying the integrity and fluidity of cell membranes and causing cell damage and resulting in autophagy [5,6]. When lens epithelial cells are continuously stimulated, and a large amount of non-functional proteins accumulate, the level of autophagy will change, resulting in a large number of lens cells being damaged and unable to perform normal physiological functions, thus forming a cataract [7,8]. The medical prevention and treatment of cataracts have gradually become an attractive research area in China and abroad in recent years.

*Buddleja officinalis* Maxim (*B. officinalis* Maxim) is a medicinal and edible plant of the *Strychnaceae* family [9,10]. The flavonoids of *B. officinalis* Maxim (BMF) are the main active ingredient of *B. officinalis* Maxim [11]. The extraction and purification of flavonoids are the premise of this plant’s medicinal function. At present, the most commonly used flavonoid extraction methods include hot water extraction, organic solvent extraction, and microwave extraction; however, the extraction rate of BMFs using these methods is not high [12,13]. The extraction rate of BMFs using enzymatic extraction was only 7.02% [12] and using microwave extraction, it was only 5.48% [13]; thus, the extraction rate of BMFs using the above methods is not high enough. In this study, a traditional extraction method, namely, heat-assisted organic solvent extraction, was used to extract BMFs. Then, the extracted flavonoids were purified using a macroporous resin to obtain flavonoids with higher purity and to detect possible monomers that play a role in its beneficial activities.

Studies have shown that BMFs can treat eye conditions. For example, Peng et al. [14] suggested that BMFs could significantly inhibit the occurrence of dry eye in rats after reducing androgen levels. Guo et al. [15] showed that BMFs had a good scavenging effect on free radicals such as ·OH, ABTS^+^, etc. However, no studies have reported on whether BMFs can resist H_2_O_2_-induced lens epithelial cell damage in rabbits by regulating oxidative stress and autophagy or on the main components of these flavonoids that aid in this role.

Therefore, this study was mainly divided into two parts: (1) extracting and purifying the BMF and analyzing its three components; (2) using the H_2_O_2_-induced rabbit lens epithelial cell injury model to explore BMFs’ protective effect on rabbit lens epithelial cell injury via the oxidative stress and autophagy pathways.

## 2. Results and Discussion

### 2.1. Analysis of the Results of the Single Factor Test

In this experiment, the extraction conditions of flavonoids were optimized by referring to Xiong et al. [16]. The main extraction methods of flavonoids include machine solvent extraction, water extraction, and microwave extraction [17,18]. The technological processes of these extraction methods are complex, have long extraction times, high energy consumption, and yield an insufficient extraction of effective components with obvious losses of organic solvents and low extraction rates. In order to overcome the shortcomings of the above technologies and increase the yield of BMFs, we used a heat-assisted organic solvent extraction method, which has been widely used in navel oranges [19], rape bee pollen [20], and other plants. Thermal-assisted organic solvent extraction is advantageous over other methods of extracting flavonoids from natural plants [21]. In this study, we studied the effects of extraction time, liquid-to-solid ratio, extraction temperature, and ethanol volume fraction on the BMF extraction rate. We found that the total flavonoids extraction rate was highest when the fixed ethanol volume was 60%, the extraction temperature was 70 °C, the extraction time was 3 h, and the solid–liquid ratio was gradually increased, especially when the solid–liquid ratio reached 1:30 (Figure 1a). Therefore, solid–liquid ratios of 1:25, 1:30, and 1:33 were selected as the optimal solid–liquid ratios. When the ratio of liquid to the material was 1:30, the volume fraction of ethanol was 70%, and the extraction temperature was 60 °C. When the extraction time was 2.5 h, the extraction rate of flavonoids reached the highest peak (Figure 1b). Therefore, 2 h, 2.5 h, and 3 h were selected as the three levels of extraction time during optimization. When the liquid–solid ratio was 1:30 (g/mL), the ethanol volume fraction was 70%, and the extraction time was 3 h, it could be seen (as shown in Figure 1c) that when the extraction temperature was 60 °C, the extraction rate of flavonoids was highest. Therefore, temperatures of 50 °C, 60 °C, and 70 °C were selected as the three levels of extraction temperature for optimization. When the liquid–solid ratio was 1:30, the extraction temperature was 70 °C, and the extraction time was 3 h, it can be observed (Figure 1d) that when the volume fraction of ethanol is 70%, the extraction effect of flavonoids is the best. Therefore, 60%, 70%, and 80% ethanol volume fractions were selected as the three levels of ethanol volume fraction for optimization.

### 2.2. Fitting the Response Surface Model

The response surface analysis test was performed according to the single factor test results whilst taking the solid-to-liquid ratio, the volume fraction of ethanol, the extraction temperature, and the extraction time into consideration as the influencing factors and taking the total flavonoid extraction rate as the response value. The test scheme is shown in Table 1.

The experimental data were analyzed using multiple regression analysis, and it was found that the relationship between the response variable (Y) and the measured value was expressed by the following quadratic polynomial equation:Y = 19.28 + 0.49X_1_ − 0.18X_2_ − 0.40X_3_ + 0.11X_4_ + 0.43X_1_X_2_ − 0.51X_1_X_3_ + 0.46X_1_X_4_ − 0.08X_2_X_3_ − 0.32X_2_X_4_ + 0.09X_3_X_4_ − 1.44X_1_^2^ − 0.22X_2_^2^ − 0.99X_3_^2^ − 0.61X_4_^2^

In the formula, X_1_ represents the solid-to-liquid ratio (g/mL); X_2_ represents the ethanol volume fraction (%); X_3_ represents the extraction temperature (°C); X_4_ represents the extraction time (h); and Y represents the extraction rate (%).

Table 2 shows the analysis of the variance of the quadratic polynomial model. From the table, we can see that both the determination coefficient R^2^ (0.9438) and the decision coefficient R_adj_^2^ (0.8876) were quite close to 1, and the difference between the two coefficients was less than 0.2, indicating that there was a high correlation between the actual value and the predicted value [22]. In addition, when the model was significant (*p* < 0.0001), X_1_, X_3_, X_1_X_2_, X_1_X_3_, X_1_X_4_, X_1_^2^, X_3_^2^, and X_4_^2^ (*p* < 0.05) were significant. The above results reflect the validity of the model, and the coefficient of variation (C.V.% = 1.85) demonstrates that the experimental data were reliable.

The predicted model is shown in Figure 2. Figure 2 shows that the interaction between any two extraction factors in this experiment was significant and that the interactions of these factors had an additive effect on the extraction efficiency of BMFs. In addition, a three-dimensional effect surface map was generated, which showed the interaction between various factors (Figure 2).

Xiong et al. [16] used an orthogonal experiment to optimize the extraction conditions of BMF, and the results showed that under the optimal condition, the extraction rate of BMF was 16.72%. In addition, Luo et al. [23] used a micro-blog extraction method to extract BMF, and the extraction rate of BMF was only 5.91%. In this study, the optimized results show that the optimal extraction parameters to obtain the highest total flavonoids yield are as follows: liquid–solid ratio of 1:31 g/mL, an ethanol volume fraction of 67%, an extraction time of 2.6 h, and an extraction temperature of 58 °C. Under optimal conditions, the average flavonoid extraction rate can reach 19.27%, which is only 0.15% lower than the predicted extraction rate. It was found that the BMF extraction process was optimized by the response surface method, and compared with the existing related reports [16], the-total flavonoid extraction rate was improved by a certain degree. In addition, the purity of the flavonoids in the sample was 76.51% after undergoing X-5-type resin purification.

### 2.3. Determination Results of Three Active Components (Luteolin, Apigenin, and Acacetin)

Up to now, research showed that the main aglycones of BMF were luteolin, apigenin, and acacetin, which had the effects of antioxidation, anti-liver injury, scavenging free radicals, and so on [23,24,25]. A previous study confirmed that apigenin has significant antioxidant, anti-inflammatory, and other activities, and it has been shown to increase intracellular GSH levels by activating oxidative stress promoters [26]. Acacetin also has pharmacological effects, such as antioxidant and antibacterial ones [27]. Luteolin was reported to be a flavonoid compound with anti-peroxidative properties [28,29]. Of course, in addition to the above three substances, there was also a small amount of linarin and verbascoside in BMF, linarin can inhibit cardiomyocyte apoptosis through PI3K/Akt/mTOR signal pathway and inhibit TLR4/Iκ Bα/NF- κB signal pathway alleviates inflammatory damage of vascular endothelial cells [30]. Additionally, verbascoside can inhibit the proliferation and migration of human glioblastoma cell lines U87 and U251, directly target c-Met to inhibit EMT, and inhibit the proliferation of human colon cancer HCT-116 cells [31]. However, in the treatment of eye diseases with BMF, researchers are still inclined to study luteolin, apigenin, and acacetin. Therefore, luteolin, apigenin, and acacetin were detected in this experiment to lay a foundation for further research. In this current research, it can be seen from Figure 3 that luteolin was the most abundant active component in the sample. The linear range of luteolin was calculated to be 31.25–1000 ppb, that of apigenin was 31.25–1000 ppb, and acacetin was 3.09–62.5 ppb. The contents of luteolin, apigenin, and acacetin in the samples were 690.85 ppb, 114.91 ppb, and 5.617 ppb, respectively. These results were consistent with the results of Jung et al. [32]. Therefore, it was speculated that the material basis of BMF’s protection of rabbit lens epithelial cells from H_2_O_2_ oxidative damage might be related to the above-mentioned components.

### 2.4. Effects of H_2_O_2_ and BMFs on the Synergistic Consequence of Cells

Guan et al. [33] found that H_2_O_2_ at a concentration of 1–10 nmol/L could promote the proliferation of human lens epithelial cells, whilst H_2_O_2_ at a concentration of more than 100 μmol/L could cause cytotoxicity and cell death. Considering the different cell species and the results of previous preliminary experiments, we selected H_2_O_2_ at a concentration of 0–800 μmol/L to conduct CCK-8 experiments. When the concentration of H_2_O_2_ was 200 μmol/L, the inhibition rate of cell survival was close to IC_50_ (Figure 4a).

Based on the literature [34] and pre-experimental results, the concentration gradient of BMFs was set from 0.05 mg/mL to 0.8 mg/mL, and the CCK-8 method was used to detect the cytotoxic effect of different concentrations of BMF on normal rabbit lens epithelial cells. The results showed that when the concentration of BMF was within 0–0.2 mg/mL, it had no significant effect on cell viability. When the concentration increased to 0.4 mg/mL, the cell viability began to be affected. Therefore, 0.05, 0.1, and 0.2 mg/mL concentrations were selected as a low-dose group, middle-dose group, and high-dose groups, respectively. According to Figure 4b, the cell survival rate of each group was detected using the CCK-8 method. The results showed that compared with the blank control group, the survival rate of cells in the H_2_O_2_ injury group was significantly decreased (*p* < 0.01). Compared with the H_2_O_2_ injury group, the survival rates of cells in the groups with low, medium, and high doses of flavonoid extract were improved (*p* < 0.01).

Using the concentrations of H_2_O_2_ and BMF determined by the above experiments to act on the rabbit lens epithelial cells, the CCK-8 method was used to detect the protective effect of different concentrations of BMF on the H_2_O_2_-induced cytotoxicity of rabbit lens epithelial cells.

In the case of cell viability, the results showed that, compared with the control group, the survival rate of cells in the H_2_O_2_ injury group was significantly decreased (*p* < 0.01). Compared with the H_2_O_2_ injury group, the survival rate of cells in the groups receiving low, medium, and high doses of flavonoids increased (*p* < 0.01), as shown in Figure 4c. The survival rates of rabbit lens cells in the three dose groups of 0.05, 0.1, and 0.2 mg/mL of BMFs were higher than that in the H_2_O_2_ injury group (*p* < 0.05), indicating that BMFs have a protective effect on the injured rabbit lens cells.

### 2.5. Results of BMFs on SOD, GSH-Px, and MDA Content and the Ability to Produce·OH in Rabbit Lens Epithelial Cells

It was reported that, after oxidative stress occurs, the normal metabolism and growth of cells will be affected. In severe cases, massive cell death and slowed or even stalled proliferation can occur [35]. Living cells contain a large number of antioxidant enzymes such as SOD, GSH-Px, etc., which have been frequently used to describe the oxidative stress state of cells [36]. GSH-Px and SOD have been shown to be enzymes with good antioxidant effects that can effectively regulate and inhibit the damage caused by oxidative stress [37]. In addition, MDA was considered to be a marker of oxidative stress, which could react with proteins and nucleic acids, causing cross-linking polymerization so that proteins could not be synthesized normally. Additionally, MDA was also highly cytotoxic and could inhibit antioxidant enzymes in the body, causing oxidation stress [38]. On the other hand,·OH is a reactive oxygen species normally formed in the body, which can cause severe damage to cellular biomolecules (such as proteins, nucleic acids, lipids, etc.) [39]. In this study, compared with the control group, the contents of SOD and GSH-Px in the H_2_O_2_ injury group were significantly decreased (*p* < 0.01), and the levels of ·OH and MDA were increased (*p* < 0.01). The contents of SOD and GSH-Px in the dosage group were significantly increased (*p* < 0.01), the content of MDA was decreased (*p* < 0.01), and the ability to generate ·OH was decreased (*p* < 0.01) (Figure 5). Therefore, after the H_2_O_2_ damage to the lens epithelial cells, the contents of SOD and GSH-Px decreased significantly, whilst the contents of ·OH and MDA increased considerably. The results showed that the balance of the antioxidant system in rabbit lens epithelial cells was disrupted when treated with H_2_O_2_. The observed decrease in the antioxidant levels in rabbit lens epithelial cells was consistent with the findings of Cao et al. [40]. After BMF treatment, the contents of SOD and GSH-Px in each group were significantly increased, and the contents of MDA and·OH groups were decreased, indicating that BMFs could protect cells by relieving oxidative stress.

### 2.6. Effect of BMFs on Autophagy of Lens Epithelial Cells after Oxidative Injury

It has been found that the possible protective mechanism of cells against oxidative stress may involve autophagy [41], so we tested the effects of BMFs on autophagy. Under normal conditions, the mCherry GFP-LC3 adenovirus can express a red fluorescent protein (mCherry) and green fluorescent protein (GFP) in cells. In addition, when mCherry merges with GFP, the results are shown in Figure 6. It could be observed that, compared with the control group, after cells were treated with H_2_O_2_, the binding process of autophagy and autolysosomes was blocked, and the autophagy flow was weakened. After GFP intervention, autophagy and autolysosomes in cells could combine smoothly, and autophagic flow increased.

## 3. Materials and Methods

### 3.1. Main Reagents and Main Instruments

*B. officinalis* Maxim (Guizhou Province, China) and Rabbit (Oryctolagus cuniculus) lens epithelial cells (He Fei Zhi Wei Biological Technology Co., Ltd., Hefei, China) were used as the main samples for this study.

#### 3.1.1. Main Reagents

Fetal Bovine Serum (Gibico Corporation, New York, NY, USA); DMEM/F_12_ medium (Gibico Corporation, USA); penicillin-streptomycin solution (100×) (Gibico Corporation, New York, NY, USA); BCA protein quantitative kit (Biyuntian Institute of Biotechnology, Shanghai, China); dimethyl sulfoxide (DMSO) (Solarbio Corporation, Beijing, China); 1.25 g/L trypsin digestion solution (Gibico Corporation, New York, NY, USA); PBS (Gibico Corporation, New York, NY, USA); 30% H_2_O_2_ (Pilot Chemical Corporation, Shanghai, China); cell counting kit-8 (CCK-8) (Dojindo Laboratories, Shanghai, China); SOD test kit (Nanjing Jiancheng Institute of Biological Engineering, Nanjing, China); malondialdehyde (MDA) (Nanjing Jiancheng Institute of Biological Engineering, Nanjing, China); Glutathione Peroxidase Enzyme-linked Immunoassay Kit (Nanjing Jiancheng Institute of Biological Engineering, Nanjing, China); X-5 type macroporous resin (Solarbio Corporation, Beijing, China); luteolin (Sigma Corporation, San Francisco, CA, USA); apigenin (Sigma Corporation, San Francisco, CA, USA); and acacetin (Sigma Corporation, San Francisco, CA, USA) were also used for this research.

#### 3.1.2. Main Instruments

The following instruments were used in this study: an ultra-clean workbench (Yida Purification Steel Structure Co., Ltd., Suzhou, China); low-temperature, high-speed centrifuge (Sigma Corporation, USA); super microplate reader (Bio-Tek Corporation, Winooski, VT, USA); inverted microscope (Nikon Corporation, Tokyo, Japan); carbon dioxide incubator (Sanyo Corporation, Osaka, Japan); ultrapure water meter (Millipore Corporation, Burlington, MA, USA); CL-32L-type High-Pressure Steam Sterilizer (ALP Corporation, Nagoya, Japan); vacuum pump (Anting Scientific Instrument Factory, Shanghai, China); constant temperature water bath (Yuhua Instrument Manufacturing Corporation, Zhengzhou, China); multi-purpose constant temperature ultrasonic extraction machine (Tian Xiang Instrument Co., Ltd., Huaian, China). In addition, high-performance liquid chromatography (HPLC, Agilent Technologies Inc., Santa Clara, CA, USA); centrifuge (Anting Scientific Instrument Factory, Shanghai, China); AB-204-N-type electronic balance (Mettler-Toledo Corporation, Shanghai, China); RE-52-type rotary evaporator (Ya Rong Biochemical Instrument Factory, Shanghai, China); and DZF-6050 vacuum drying oven (Qixin Scientific Instrument Co., Ltd., Shanghai, China) were also used.

### 3.2. Methods

#### Preparation of Standard Flavonoid Products

Dry standard Rutin solution was prepared with 60% ethanol solution (0.20 mg/mL); samples of 0.0 mL, 1.0 mL, 2.0 mL, 3.0 mL, 4.0 mL, 5.0 mL, and 6.0 mL of Rutin standard solution were placed in seven 25 mL colorimetric tubes, and 1.0 mL of 5% sodium nitrite solution was then added. The tubes were left at room temperature (25 °C) for 6 min; 1.0 mL of 5% aluminum nitrate solution and 10.0 mL of 4% hydrogen-dissolved sodium oxide were then added. Additionally, 30% ethanol solution was added to 25 mL, shaken well, and left for 15 min. The absorbance was measured at 510 nm for the standard curve.

A 5.0 mL sample solution was pipetted and measured accurately and put in a 25.00 mL colorimetric tube, where the reagents were added according to Section 3.1.1. The absorbances were measured at 510 nm. The contents were calculated according to the standard curve, and the flavonoid extraction rate was calculated as per Equation (1):(1)$$Y=\frac{M_{1}\times V_{1}}{M_{2}\times V_{2}\times 10^{3}}\times 100\%$$

In the formula, *Y* represents the extraction rate (%) of BMF, *M*_1_ represents the flavonoids (mg) in the liquid to be tested calculated according to the standard curve, $M_{2}$ represents the sample mass (g), $V_{1}$ represents the volume (mL) of the sample extract solution, and $V_{2}$ represents the total volume of the sample extraction (mL).

### 3.3. Optimization of the Crude BMF Extraction Conditions

#### 3.3.1. Single Factor Test

The ethanol volume fraction, solid–liquid ratio, extraction temperature, and extraction time were selected as the influencing factors of BMF, and therefore, a single-factor experiment was carried out. The factors and levels are as follows: solid-to-liquid ratio (g/mL): 1:20, 1:25, 1:30, 1:35, 1:40; ethanol volume fraction (%): 40, 50, 60, 70, 80; extraction temperature (°C): 40, 50, 60, 70, 80; extraction time (h): 2, 2.5, 3, 3.5, 4.

#### 3.3.2. Response Surface Test Design

According to the single-factor test results, we took ethanol volume fraction, solid–liquid ratio, extraction temperature, extraction time, and other influencing factors as independent variables. The Box–Behnken design method was used to optimize these factors, and the extraction rate of total flavonoids was used as the response index to carry out a response surface optimization experiment. The factors and levels are summarized in Table 3.

### 3.4. Purification of Crude BMF

Referring to the method of Liu et al. [42], an appropriate amount of X-5 type resin was weighed, soaked in an anhydrous ethanol solution for 24 h, and then rinsed with pure water until there was no ethanol smell detected. Then, it was soaked in 5% NaOH solution for 6 h, rinsed with pure water to wash until effluent became neutral, and finally soaked in 5% HCl solution for 6 h and rinsed with pure water again to make it neutral. Later, the pretreated X-5 resin was packed into a column. The sample loading mass concentration was 6.00 mg/mL, the loading flow rate was 2 BV/h, the loading volume was 20 BV, and the desorption flow rate was 1.5 BV/h; 80% ethanol was used for desorption. The desorption solution was collected; then, the desorption solutions were decompressed to recover ethanol until no droplets appeared. Finally, it was placed in a 4 °C refrigerator for later use.

### 3.5. Determination of Luteolin, Apigenin, and Acacetin

#### 3.5.1. Standard Product Solution Preparation

Accurately measured 1.00 ± 0.1 mg samples of acacetin, apigenin, and luteolin reference substances were weighed, placed in a 15.00 mL centrifuge tube, and 1 mL of ethanol was added. Next, 10 µL of the above solution was placed into a 10.00 mL volumetric flask, and ultra-pure water was added to fix the volume to 10.00 mL solution. Then, standard stock solutions were prepared with varying concentrations: luteolin: 31.25 ppb, 62.5 ppb, 125 ppb, 250 ppb, 500 ppb, 1000 ppb; apigenin: 31.25 ppb, 62.5 ppb, 125 ppb, 250 ppb, 500 ppb, 1000 ppb; acacetin: 3.09 ppb, 7.81 ppb, 15.625 ppb, 31.25 ppb, 62.5 ppb.

#### 3.5.2. Preparation of the BMF Sample Fluid

Extraction of the BMF solutions was carried out according to the results of the response surface test design, and the crude extract was purified according to Section 3.3.1.

#### 3.5.3. HPLC Detection Conditions

The chromatographic column used was a Waters Sun Fire TM C_18_ (4.6 mm × 250 mm, 5 µm). The mobile phase consisted of acetonitrile, 0.2% aqueous phosphoric acid gradient elution was performed, the flow rate was 1.0 mL/min, the column temperature was 30 °C, the wavelength was 330 nm, and the injection volume was 10 µL. The gradient elution procedure is shown in Table 4.

### 3.6. Cell-Related Processing and Experiments

After obtaining the rabbit lens epithelial cells, we digested them with 1 mL of 0.25% trypsin and placed them in a 37 °C, 5% CO_2_ incubator for 3 min; 1.0 mL of culture solution was drawn to terminate the digestion. We performed 1000 rpm centrifugation for 5 min, selecting an appropriate ratio for subculture according to cell growth, and 3–5 generations of rabbit lens epithelial cells were selected for this experiment.

#### 3.6.1. Determination of the Half-Inhibitory Concentration of H_2_O_2_ on the Survival of Rabbit Lens Epithelial Cells by CCK-8 Method

Cultures of 1 × 10^4^ mL^−1^ of cells were inoculated in each well of a 96-well plate and cultured for 24 h. They were divided into the following groups (where there are 5 duplicate holes in each group):

Test groups: different concentrations (0, 50, 100, 200, 400, 800 μmoL/L) of H_2_O_2_ added;

Control group: the same volume of complete medium and cells as the experimental group was added;

Blank group: the same volume of the complete medium as the experimental group was added.

After culturing for 24 h in a 37 °C, 5% CO_2_ incubator, 10 μL of CCK-8 solution was added to each well and left to incubate for 2 h. The absorbance (A) value of each well was measured with a microplate reader at a wavelength of 450 nm. The experiment was repeated 3 times. The cell survival rate and cell survival inhibition rate to obtain IC_50_ was calculated.

#### 3.6.2. CCK-8 Method to Determine the Half-Inhibitory Concentration of BMFs on the Survival of Rabbit Lens Epithelial Cells

In each well, cells were inoculated at a cell density of 1 × 10^4^ mL^−1^ in a 96-well plate and cultured for 24 h. They were divided into the following groups (there are 5 duplicate holes in each group):

Experimental groups: different concentrations (0, 0.05, 0.1, 0.2, 0.4, and 0.8 mg/mL) of BMFs added;

Control group: the same volume of complete medium and cells as the experimental groups were added.

Blank group: the same volume of the complete medium as the experimental group was added.

After culturing at 37 °C in a 5% CO_2_ incubator for 24 h, 10 μL of CCK-8 solution was added to each well. Later, the incubation was continued for 2 h, and the values of each well were measured on a microplate reader at a wavelength of 450 nm. The experiment was repeated three times. Then, the cell survival rate and cell survival inhibition rate to obtain IC_50_ were calculated using Equation (2):(2)$$\mathrm{Cell}\;\mathrm{survival}\;\mathrm{rate}\;\left(\%\right)=\frac{\mathrm{As}-\mathrm{Ab}}{\mathrm{Ac}-\mathrm{Ab}}\times 100\%$$
(3)Cell survival inhibition rate (%)=1−cell survival rate (%)
where As is the average value of the experimental group, Ab is the average value of the blank group, and Ac is the average value of the control group.

#### 3.6.3. CCK-8 Assay to Detect the Protective Effect of BMF on H_2_O_2_-Induced Cytotoxicity of Rabbit Lens Epithelial Cells

Cells were inoculated in 96-well plates at a cell density of 1 × 10^4^ mL^−1^ and cultured for 24 h. They were divided into the following groups (there are 5 duplicate holes in each group):

Blank control group: rabbit lens epithelial cells + DMEM/F12 medium;

H_2_O_2_ injury group: blank control group + 200 μmol/L H_2_O_2;_

Low-dose group: H_2_O_2_ injury group + 0.05 mg/mL flavonoids extract;

Middle-dose group: H_2_O_2_ injury group + 0.1 mg/mL flavonoids extract;

High-dose group: H_2_O_2_ injury group + 0.2 mg/mL flavonoids extract.

An amount of 200 μmol/L H_2_O_2_ was added to normal cultured rabbit lens epithelial cells and incubated for 24 h, followed by adding 0.05, 0.1, and 0.2 mg/mL of BMFs for 24 h, adding 10 μL CCK-8 solution to each well, continuing to incubate for 2 h, and loading on the microplate reader. The absorbance was measured at a wavelength of 450 nm. The experiment was repeated three times. Calculation of the cell survival rate and cell survival inhibition rate to obtain IC_50_ was conducted using Equations (4) and (5), respectively.
(4)$$\mathrm{Survival}\;\mathrm{inhibition}\;\mathrm{rate}\;\left(\%\right)=\frac{\mathrm{OD}\;\mathrm{value}\;\mathrm{of}\;\mathrm{control}\;\mathrm{group}\;-\;\mathrm{OD}\;\mathrm{value}\;\mathrm{of}\;\mathrm{drug}\;\mathrm{group}}{\mathrm{OD}\;\mathrm{value}\;\mathrm{of}\;\mathrm{the}\;\mathrm{control}\;\mathrm{group}}\times 100\%$$
(5)Survival rate (%)=1−Survival inhibition rate (%)

### 3.7. Detection of SOD, GSH-Px, and MDA Content and the Ability to Produce·OH in Rabbit Lens Epithelial Cells

#### 3.7.1. Preprocessing of Cells

After grouping the cells according to Section 3.5.3, adherent cells were collected and centrifuged at 1000 rpm for 10 min, and the supernatant fluid was discarded. Before the measurement, a quantitative buffer (PBS) was added; then, the cells were processed by sonication. The sonication was performed at 300 W, with sonication every 5 s, 4 times. Later, further centrifugation was performed to collect the supernatant for subsequent experiments.

#### 3.7.2. Detection of SOD Activity

The SOD Activities Were Detected According to the SOD Kit Instructions (Equation (6))
(6)$$\mathrm{Total}\;\mathrm{SOD}\;\mathrm{activity}\;\left(\frac{U}{\mathrm{mgprot}}\right)=\frac{\frac{1}{2}\mathrm{SOD}\;\mathrm{inhibition}\;\mathrm{rate}}{\mathrm{Protein}\;\mathrm{concentration}\;\mathrm{of}\;\mathrm{the}\;\mathrm{samples}\;\mathrm{to}\;\mathrm{be}\;\mathrm{tested}\;\left(\mathrm{mgprot}/\mathrm{mL}\right)}$$

#### 3.7.3. Detection of MDA Content

The MDA Content Was Carried Out as per Equation (7)

The test was performed according to the *MDA* kit instructions.
(7)$$MDA\;\mathrm{content}=\frac{\mathrm{measured}\;\mathrm{OD}\;\mathrm{value}\;-\;\mathrm{control}\;\mathrm{OD}\;\mathrm{value}}{\mathrm{standard}\;\mathrm{OD}\;\mathrm{value}\;-\;\mathrm{blank}\;\mathrm{OD}\;\mathrm{value}}\times \frac{\mathrm{standard}\;\mathrm{concentration}}{\;\mathrm{sample}\;\mathrm{protein}\;\mathrm{concentration}}$$

#### 3.7.4. Detection of GSH-Px Activity

GSH-Px Activities Were Also Detected (Equation (8))
(8)$$\begin{array}{l} GSH-Px\;\mathrm{enzyme}\;\mathrm{activity}:\\ =\frac{\mathrm{OD}\;\mathrm{value}\;\mathrm{of}\;\mathrm{non}-\mathrm{enzyme}\;\mathrm{tube}\;-\;\mathrm{OD}\;\mathrm{value}\;\mathrm{of}\;\mathrm{enzyme}\;\mathrm{tube}}{\mathrm{OD}\;\mathrm{value}\;\mathrm{of}\;\mathrm{standard}\;\mathrm{tube}\;-\;\mathrm{OD}\;\mathrm{value}\;\mathrm{of}\;\mathrm{blank}\;\mathrm{tube})}\\ \times \frac{\mathrm{the}\;\mathrm{concentration}\;\mathrm{of}\;\mathrm{standard}\;\mathrm{tube}\;\left(20\;\mathrm{umol}/L\right)\times \;\mathrm{dilution}\;\mathrm{ratio}}{\;\mathrm{reaction}\;\mathrm{time}\times \mathrm{sample}\;\mathrm{amount}\;\times \;\mathrm{sample}\;\mathrm{protein}\;\mathrm{content}} \end{array}$$

#### 3.7.5. Generate OH Group

Generate OH Group Is Performed by Adopting Equation (9)
(9)$$\begin{array}{l} \mathrm{Ability}\;\mathrm{to}\;\mathrm{generate}\;-OH\;\left(U/\mathrm{mL}\right)\\ =\frac{\mathrm{measured}\;\mathrm{OD}\;\mathrm{value}\;-\;\mathrm{control}\;\mathrm{OD}\;\mathrm{value}}{\mathrm{standard}\;\mathrm{OD}\;\mathrm{value}\;-\;\mathrm{blank}\;\mathrm{OD}\;\mathrm{value}}\\ \times \frac{\mathrm{standard}\;\mathrm{concentration}\times \mathrm{sample}\;\mathrm{dilution}\;\mathrm{concentration}\;\mathrm{before}\;\mathrm{testing}}{\mathrm{sample}\;\mathrm{volume}} \end{array}$$

### 3.8. Transfection of mRFP-GFP-LC3 Tandem Fluorescent Proteins into Cells by Adenovirus

The cells were evenly seeded into 24-well plates at concentrations of 1 × 10^5^ mL^−1^ cells according to Section 3.5.3. After 24 h, the cells grew to 50–70%. Then, 500 μL of BMFs were added to each well, and the virus (10^7^ mL^−1^) was added after 24 h. After 24 h, the cells were photographed and analyzed using a fluorescence microscope.

### 3.9. Detection of Autophagy Marker Protein Expression in Rabbit Lens Epithelial Cells by Western Blotting

A 6-well plate was used to inoculate rabbit lens epithelial cells, as described in Section 2.4. After grouping and processing, the supernatant was discarded, washed twice using PBS, and then lysed with RIPA buffer. The BCA protein assay was used to analyze the protein concentration. The equivalent protein (20 μL) was electrophoresed on a PAGE gel (12%), and the electrophoresis condition was 80 V for 2 h. After electrophoresis, it was transferred to a polyvinylidene fluoride (PVDF) membrane. Then, the membrane was blocked with 5% condensed milk (fat-free) in a water bath at 37 °C for 30 min. After washing three times with TBST, membranes were incubated at 4 °C with primary antibodies from P62 (1:2000, mouse), Beclin-1 (1:2000, mouse), and GAPDH (1:2000, mouse) for 10 h. After washing three times with TBST, the membrane was incubated with a secondary antibody (1:5000, goat) for 2 h at room temperature. Finally, protein expression was examined by using an enhanced chemiluminescence (ECL) kit. GAPDH was used as the control protein. All the results were evaluated by grayscale analysis using the ImageJ software (National Institutes of Health, Bethesda, MD, USA.

### 3.10. Data Analysis

The result analysis software used in this experiment was Design Expert 10, Graph Pad Prism 8, ImageJ, and IBM SPSS Statistics 22 (International Business Machines Corporation, New York, NY, USA). Statistical significance was established as *p* < 0.05. All the measurements were performed three times. The mean ± standard deviation (SD) was used to express the results.

## 4. Conclusions

In this research, after optimizing the extraction conditions of BMFs, it was found that when the liquid–solid ratio was 1:31 g/mL, the volume fraction of ethanol was 67%, the extraction time was 2.6 h, and the extraction temperature was 58 °C; the BMF extraction rate could reach 19.27%. In addition, the three main monomers of BMFs were detected by HPLC, including luteolin, apigenin, and acacetin. The results of this study also indicated that the material basis of BMFs’ resistance to H_2_O_2_-induced lens cell injury in rabbits might be related to the content of glycosides. BMFs have a significant protective effect on H_2_O_2_-induced injuries of rabbit lens epithelial cells, and the mechanism of this effect may be related to improving intracellular redox homeostasis and the autophagy pathway. Therefore, starting from the above three monomers, as well as oxidative stress and autophagy, we can study the specific components and mechanisms of BMF against oxidative damage in rabbit lens epithelial cells next.

## Figures and Tables

**Figure 1 molecules-27-08985-f001:**
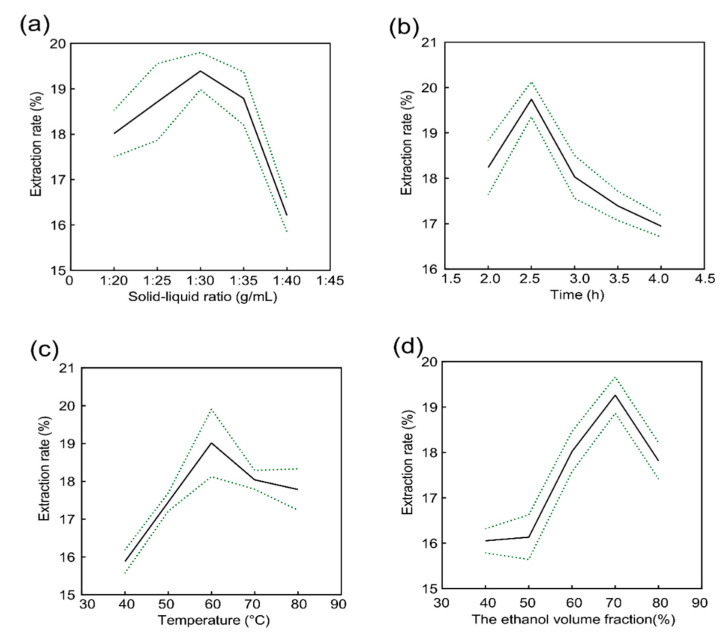
One-factor experiment results: (**a**) Effect of solid-to-liquid ratio on the extraction rate of BMF; (**b**) Effect of time on the extraction rate of BMF; (**c**) Effect of temperature on the extraction rate of BMF; (**d**) Effect of ethanol volume fraction on the extraction rate of BMF. The solid line represents the mean, and the dotted line represents the error line (Each value was represented as the mean ± SD of three independent experiments, n = 3).

**Figure 2 molecules-27-08985-f002:**
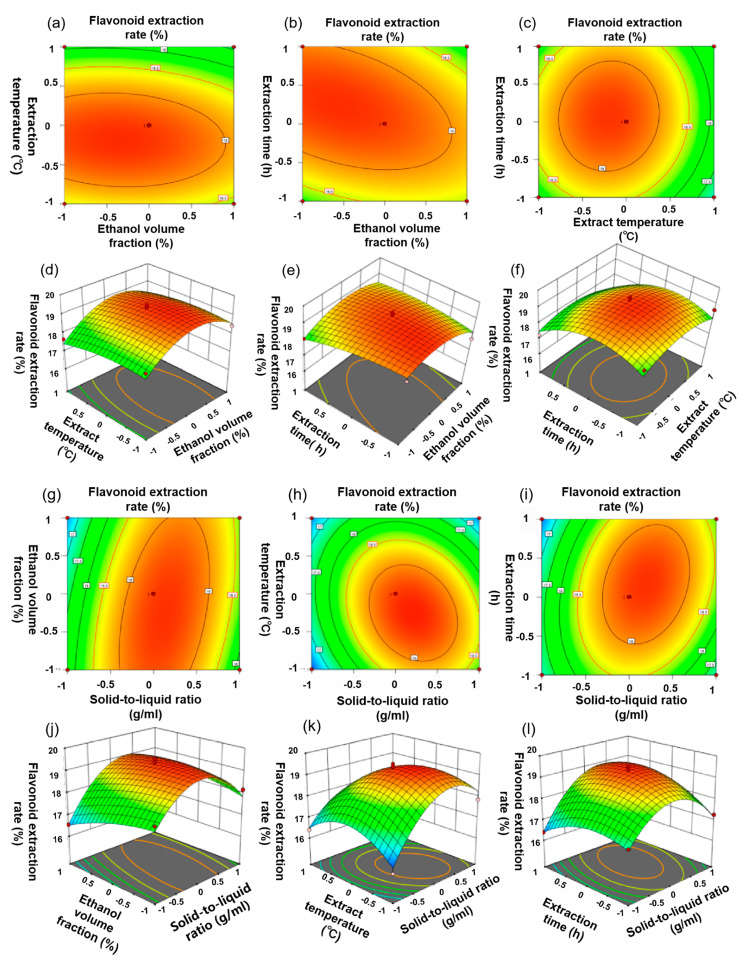
(**a**–**c**,**g**–**i**) represent the isoline map with significant interaction of extraction parameters; (**d**–**f**,**j**–**l**) represents the interaction of four factors affecting the extraction rate of BMF using response surface methodology.

**Figure 3 molecules-27-08985-f003:**
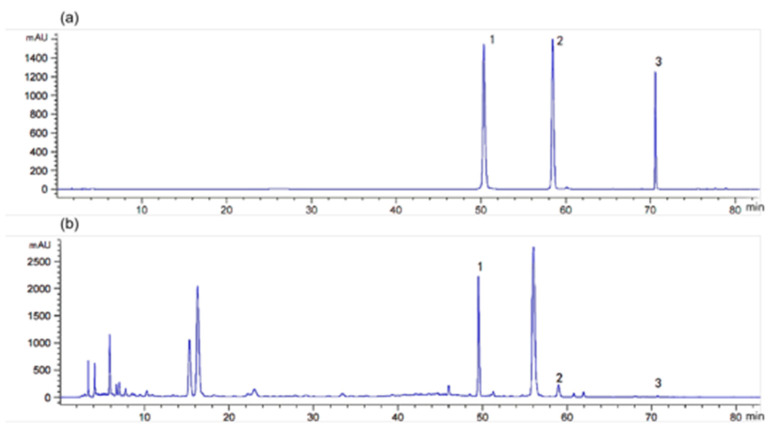
Determination of flavonoids (luteolin, apigenin, acacetin) in standard and sample by HPLC: (**a**) HPLC of standards; (**b**) HPLC of the samples. 1 represents luteolin, 2 represents apigenin, 3 represents acacetin (Appendix A).

**Figure 4 molecules-27-08985-f004:**
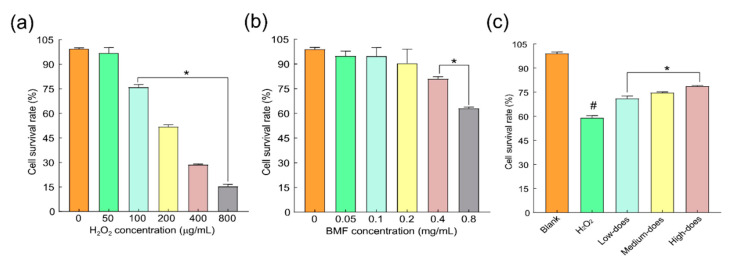
Effects of different concentrations of H_2_O_2_ and flavonoids on the viability of rabbit lens epithelial cells. (**a**) Effects of BMF concentration on cell survival rate; (**b**) Effects of H_2_O_2_ concentration on cell survival rate; (**c**) Effects of BMFs in different doses on the oxidative damage due to H_2_O_2_. # Symbolizes compared with blank group, *p* < 0.01; * symbolizes compared with H_2_O_2_ injury group, *p* < 0.01.

**Figure 5 molecules-27-08985-f005:**
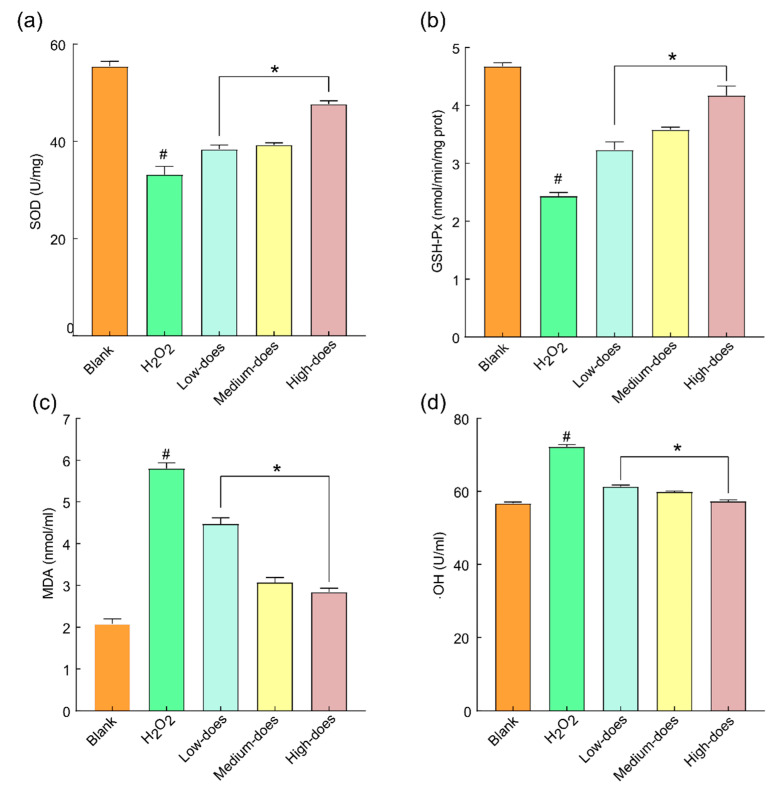
Effects of BMFs on SOD, GSH-Px, MDA, and ·OH in H_2_O_2_-damaged rabbit lens epithelial cells: (**a**) Effects of BMFs on SOD in H_2_O_2_-damaged rabbit lens epithelial cells; (**b**) Effects of BMFs on GSH-Px in H_2_O_2_-damaged rabbit lens epithelial cells; (**c**) Effects of BMF on MDA in H_2_O_2_-damaged rabbit lens epithelial cells; (**d**) Effects of BMFs on S·OH in H_2_O_2_-damaged rabbit lens epithelial cells. * Symbolized compared H_2_O_2_ injury group: *p* < 0.01; # symbolized compared blank control group: *p* < 0.01.

**Figure 6 molecules-27-08985-f006:**
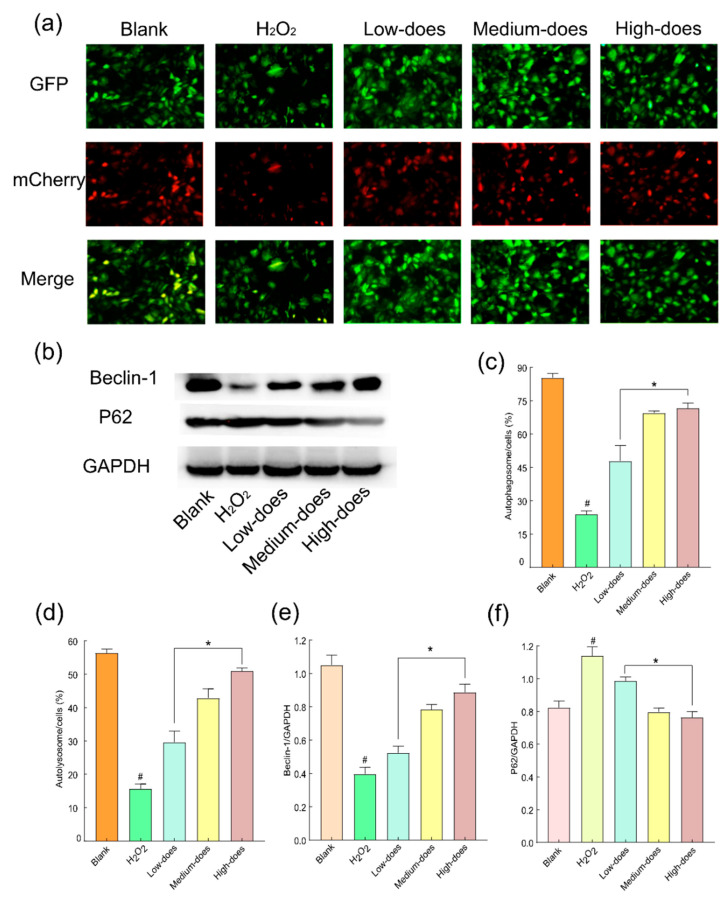
Effect of BMFs on the autophagy of rabbit lens epithelial cells after oxidative injury: (**a**,**c**,**d**) Effect of BMFs on autophagic flow after oxidative damage in rabbit lens epithelial cells; (**b**,**e**,**f**) Effect of BMFs on autophagy-related proteins after oxidative damage in rabbit lens epithelial cells. # Symbolizes compared with blank group, *p* < 0.01; * symbolizes compared with H_2_O_2_ injury group, *p* < 0.01.

**Table 1 molecules-27-08985-t001:** Response surface test design.

Test Number	(X_1_) Solid-to-Liquid Ratio (g/mL)	(X_2_) The Ethanol Volume Fraction (%)	(X_3_) Extract Temperature (°C)	(X_4_) Extract Time (h)	(Y) Extraction Rate (%)
1	0	−1	0	1	18.84 ± 0.52
2	0	−1	0	−1	17.73 ± 0.20
3	1	0	−1	0	17.86 ± 0.88
4	0	0	−1	−1	18.61 ± 0.33
5	0	1	0	−1	18.25 ± 0.57
6	−1	1	0	0	16.55 ± 0.37
7	−1	0	1	0	16.40 ± 0.11
8	0	0	0	0	19.51 ± 0.63
9	1	0	0	1	18.32 ± 0.50
10	0	−1	1	0	18.21 ± 0.43
11	0	0	−1	1	18.48 ± 0.71
12	0	0	1	1	17.25 ± 0.82
13	0	0	0	0	19.24 ± 0.82
14	−1	0	−1	0	16.00 ± 0.68
15	1	0	1	0	16.24 ± 0.93
16	0	1	0	1	18.07 ± 0.67
17	1	−1	0	0	18.16 ± 0.26
18	1	1	0	0	18.48 ± 0.19
19	0	0	1	−1	17.03 ± 0.84
20	0	0	0	0	19.11 ± 0.18
21	1	0	0	−1	17.25 ± 0.63
22	0	1	1	0	17.66 ± 0.94
23	−1	−1	0	0	17.96 ± 0.82
24	0	0	0	0	19.17 ± 0.37
25	−1	0	0	1	16.40 ± 0.09
26	0	−1	−1	0	18.43 ± 0.73
27	0	0	0	0	19.38 ± 0.39
28	−1	0	0	−1	17.18 ± 0.44
29	0	1	−1	0	18.21 ± 0.72

Note: X_1_ represents the solid-to-liquid ratio (g/mL); X_2_ represents the ethanol volume fraction (%); X_3_ represents extraction temperature (°C); X_4_ represents extraction time (h); Y represents extraction rate (%).

**Table 2 molecules-27-08985-t002:** Response Surface Quadratic Model Variance Analysis and regression coefficient.

Source	Sum of Squares	DF	Mean Square	F-Value	*p*-Value	Statistical Difference
Model	25.91	14	1.85	16.79	<0.0001	**
X_1_	2.82	1	2.82	25.6	0.0002	**
X_2_	0.371	1	0.371	3.36	0.0879	ns
X_3_	1.92	1	1.92	17.41	0.0009	**
X_4_	0.143	1	0.143	1.3	0.2739	ns
X_1_X_2_	0.7482	1	0.7482	6.79	0.0208	*
X_1_X_3_	1.02	1	1.02	9.25	0.0088	**
X_1_X_4_	0.8556	1	0.8556	7.76	0.0146	*
X_2_X_3_	0.0272	1	0.0272	0.2469	0.627	ns
X_2_X_4_	0.416	1	0.416	3.77	0.0725	ns
X_3_X_4_	0.0306	1	0.0306	0.2777	0.6064	ns
X_1_^2^	13.45	1	13.45	121.94	<0.0001	**
X_2_^2^	0.3168	1	0.3168	2.87	0.1122	ns
X_3_^2^	6.39	1	6.39	57.92	<0.0001	**
X_4_^2^	2.44	1	2.44	22.14	0.0003	**
Residual	1.54	14	0.1103			
Lack of Fit	1.44	10	0.1438	5.45	0.0582	not significant
Pure Error	0.1055	4	0.0264			
Cor Total	27.46	28				
C.V.%	1.85					
R^2^	0.9438					
R_adj_^2^	0.8876					

Note: X_1_ represents the solid-to-liquid ratio (g/mL); X_2_ represents the ethanol volume fraction (%); X_3_ represents extraction temperature (°C); X_4_ represents extraction time (h); * represents significant difference (0.01 < *p* < 0.05); ** represents extreme significant difference (*p* < 0.01).

**Table 3 molecules-27-08985-t003:** Factor levels of response surface analysis.

Factors	Levels		
	−1	0	1
Solid-to-liquid ratio (g/mL)	25	30	35
The ethanol volume fraction (%)	60	70	80
Extraction temperature (°C)	50	60	70
Extraction time (h)	2	2.5	3

**Table 4 molecules-27-08985-t004:** Gradient elution procedure.

Time (min)	Acetonitrile (%)	0.2% Phosphoric Acid an Aqueous Solution (%)
0–3	18	82
3–25	18→19	82→81
25–32	19→20	81→80
32–60	20→38	80→62
60–61	38→39	62→61
61–62	39→40	61→60
62–75	40→100	60→0
75–85	100	0

## Data Availability

Not applicable.

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
