# Peer review of "Extraction and Purification of Flavonoids from Buddleja officinalis Maxim and Their Attenuation of H2O2-Induced Cell Injury by Modulating Oxidative Stress and Autophagy"

_molecules, 2022, doi:10.3390/molecules27248985_

Round 1
Reviewer 1 Report
The authors in this study optimized the BMF extraction and detected three main active monomers (Luteolin, Apigenin, and Acacetin). They evaluate the protective levels of the BMF fraction in a standard oxidative damage model of rabbit lens epithelial cells using hydrogen peroxide (H2O2) as an insulant. In their in vitro assay, they detected the levels of superoxide dis-22 mutase, glutathione peroxidase, malondialdehyde and OH radical. The expression of Beclin-1 and P62 were detected using Western Blot experiments. The results suggest that BMF increases the levels of superoxide dismutase and glutathione peroxidase and decreases the levels of malondialdehyde. Additionally, the author claims that BMF could promote the binding of autophagosomes to lysosomes.
Overall, the current study lacks novelty and additional experiments to support the mechanism of action of BMF. Therefore, this work in its current version is not suitable for the journal.
Author Response
Thank you for your valuable comments. And please allow me to give this explanation as explained below:
Oxidative stress is one of the main factors influencing the pathogenesis of cataract. Autophagy is a physiological phenomenon with self-protection function in eukaryotic cells. It mainly helps cells to adapt to various adverse stimuli, and it is the basic way for cells to maintain their internal environmental self-stability and achieve self-renewal. By reviewing the relevant literature, we found no reports of BMF extracted from Buddleja officinalis in H2O2-induced oxidative stress by regulating the level of autophagy. That is why we started this research and hoping to get your understanding and support. Furthermore, as for mechanistic studies, in this paper, we plan to first understand whether the BMF we extracted has the above role, and the mechanisms will be investigated next if possible.
I hope you can consider our manuscript again. Thank you very much.
Best wishes,
Shaofeng Wei
Reviewer 2 Report
The present research is of interest for Molecules in my opinion, however I recommend a major revision, taking into account the following points:
- write luteolin, apigenin, and acacetin instead of Luteolin, Apigenin, and Acacetin.
- use B. officinalis, after the first use of the complete name Buddleja officinalis
- Figure 1: include error bars
- Table 1: add SD or SE for extraction rate %
- Provide LOD, LOQ, standard curves and R2 for each phytochemical.
- Compare the concentration obtained in luteolin, apigenin, and acacetin with litterature data from the same plant species and with other plant species sources, to discuss the interest of the extraction methods and the present plant source for the extraction of these compounds.
- Figure 4: the flavonoid concentrations described in the text and those presented in Figure 4 are not in the same unit. Please correct.
- I do not understand the differences in survival observed between Figure 4a and 4b for the high concentration. Could you discuss this result a bit more.
Author Response
The present research is of interest for Molecules in my opinion, however I recommend a major revision, taking into account the following points:
Response: Thanks for handling our manuscript and providing the opportunity for improving further. We have tried our best to work on every comments to improve the manuscript. We have responded them point by point marked through track change options throughout the manuscript to trace them easily.
Moreover,
a)We have cross-checked the English language by English Editing from MDPI to proofread our manuscript.
We used Microsoft Track change option to trace all the changes easily.
- write luteolin, apigenin, and acacetin instead of Luteolin, Apigenin, and Acacetin.
Response: Thanks for your reminder, we have modified according to your opinion.
- use B. officinalis, after the first use of the complete name Buddleja officinalis
Response: Thank you very much, we have modified it according to your comment.
- Figure 1: include error bars
Response: Thanks very much, the error bars in Figure 1 are represented in dashed lines. thanks again.
- Table 1: add SD or SE for extraction rate %
Response: In Table 1, we have added the SD. Hope to learn some more from you.
- Provide LOD, LOQ, standard curves and R2 for each phytochemical.
Response: Thank you for your patient inspection. We have provided LOD, LOQ, the standard curve formula and R2 in the supplemental material on the final page of the manuscript.
- Compare the concentration obtained in luteolin, apigenin, and acacetin with litterature data from the same plant species and with other plant species sources, to discuss the interest of the extraction methods and the present plant source for the extraction of these compounds.
Response: Thank you for your reminding. The influence of different extraction methods on the extraction rate of BMF has been compared in the manuscript
- Figure 4: the flavonoid concentrations described in the text and those presented in Figure 4 are not in the same unit. Please correct.
- I do not understand the differences in survival observed between Figure 4a and 4b for the high concentration. Could you discuss this result a bit more.
Response: Figure 4a showed the cell survival rate under different concentrations of H2O2. When H2O2 was 200 μmol/L, the inhibition rate of cell survival was close to IC50. Therefore, 200 μmol/L was determined as the optimal concentration of H2O2. Figure 4b showed the survival rate of cells under the intervention of different concentrations of BMF (because excessive BMF will cause certain damage to cells, while too low concentration will lead to no effect, so we need to make sure that the cells have a certain response, but the concentration of BMF was not enough to harm cells). Thank you for your time and effort for our manuscript.
Reviewer 3 Report
This is an interesting manuscript, but I have a few observations:
1. There are some writing errors (line 99) and writing (line 106) that make reading and understanding difficult.
2. References must be written according to the journal's standards and review.
3. Rabbit lens epithelial cells, were they purchased? Does your production conform to ethical standards? I ask these questions because a certificate that endorses bioethical compliance in animal handling is not indicated or presented.
4. Species of the genus Buddleja, show an interesting pool of compounds; the authors only use three of them, but it is not clear why use only these three.
Author Response
This is an interesting manuscript, but I have a few observations:
Response: Thanks for handling our manuscript and forwarding the review comments. We have tried our best to work on every comments to improve the manuscript. We have responded them point by point marked through track change options throughout the manuscript to trace them easily.
Moreover,
- We have cross-checked the English language by MDPI to proofread our manuscript.
- We used Microsoft Track change option to trace all the changes easily.
1. There are some writing errors (line 99) and writing (line 106) that make reading and understanding difficult.
Response: Thanks for your reminder, we have modified according to your opinion.
2. References must be written according to the journal's standards and review.
Response: Thank you very much. We have modified the format of references
3. Rabbit lens epithelial cells, were they purchased? Does your production conform to ethical standards? I ask these questions because a certificate that endorses bioethical compliance in animal handling is not indicated or presented.
Response: Our cells were purchased from Hefei Hesheng Biotechnology Co., Ltd. They provided the quality certificate of experimental animals. Therefore, our research is in line with ethical standards. Thank you for your time and effort for our manuscript.
4. Species of the genus Buddleja, show an interesting pool of compounds; the authors only use three of them, but it is not clear why use only these three.
Response: Luteolin and other flavonoids are the three most widely studied and more representative substances, so we also made the hypothesis that these three substances may be the material basis for the effect of BMF. If possible, our next research will further explore the relationship between these three monomers and BMF activity based on this current research. Thank you again.
Round 2
Reviewer 2 Report
The Authors have answered in a clear way to all my questions about the first version.
Author Response
None
Reviewer 3 Report
Thanks for the answers, I have attached an article link that may be interesting: Molecules 2021, 26, 2192. https://doi.org/10.3390/molecules26082192
Author Response
none